# SSIF: Learning Continuous Image Representation for Spatial-Spectral Super-Resolution

## Abstract

Existing digital sensors capture images at fixed spatial and spectral resolutions (e.g., RGB, multispectral, and hyperspectral images), and each combination requires bespoke machine learning models. Neural Implicit Functions partially overcome the spatial resolution challenge by representing an image in a resolution-independent way. However, they still operate at fixed, pre-defined spectral resolutions. To address this challenge, we propose Spatial-Spectral Implicit Function (SSIF), a neural implicit model that represents an image as a function of both continuous pixel coordinates in the spatial domain and continuous wavelengths in the spectral domain. We empirically demonstrate the effectiveness of SSIF on two challenging spatio-spectral super-resolution benchmarks. We observe that SSIF consistently outperforms state-of-the-art baselines even when the baselines are allowed to train separate models at each spectral resolution. We show that SSIF generalizes well to both unseen spatial resolutions and spectral resolutions. Moreover, SSIF can generate high-resolution images that improve the performance of downstream tasks (e.g., land use classification) by 1.7%-7%.

## 1 Introduction

While the physical world is continuous, most digital sensors (e.g., cell phone cameras, multispectral or hyperspectral sensors in satellites) can only capture a discrete representation of continuous signals in both spatial and spectral domains (i.e., with a fixed number of spectral bands, such as red, green, and blue). In fact, due to the limited energy of incident photons, fundamental limitations in achievable signal-to-noise ratios (SNR), and time constraints, there is always a trade-off between spatial and spectral resolution (Mei et al., 2020; Ma et al., 2021)[1]. High spatial resolution and high spectral resolution can not be achieved at the same time, leading to a variety of spatial and spectral resolutions used in practice for different sensors. However, ML models are typically bespoke to certain resolutions, and models typically do not generalize to spatial or spectral resolutions they have not been trained on. This calls for image super-resolution methods.

The goal of image super-resolution (SR) (Ledig et al., 2017; Lim et al., 2017; Zhang et al., 2018b; Haris et al., 2018; Zhang et al., 2020c; Yao et al., 2020; Mei et al., 2020; Saharia et al., 2021; Ma et al., 2021; He et al., 2021) is to increase the spatial or spectral resolution of a given single low-resolution image (Galliani et al., 2017). It has become increasingly important for a wide range of tasks including object recognition and tracking (Pan et al., 2003; Uzair et al., 2015; Xiong et al., 2020), medical image processing (Lu & Fei, 2014; Johnson et al., 2007), remote sensing (He et al., 2021; Bioucas-Dias et al., 2013; Melgani & Bruzzone, 2004; Zhong et al., 2018; Wang et al., 2022a) and astronomy (Ball et al., 2019).

Traditionally image SR has been classified into three tasks according to the input and output image resolutions:[2] Spatial Super-Resolution (spatial SR), Spectral Super-Resolution (spectral SR) and Spatio-Spectral Super-Resolution (SSSR). Spatial SR (Zhang et al., 2018a; Hu et al., 2019; Zhang et al., 2020a; Niu et al., 2020; Wu et al., 2021b; Chen et al., 2021; He et al., 2021) focuses on

---

[1] Given a fixed overall sensor size and exposure time, higher spatial resolution and higher spectral resolution require the per pixel sensor to be smaller and bigger at the same time, which are contradicting each other.

[2] A related task, Multispectral and Hyperspectral Image Fusion (Zhang et al., 2020c; Yao et al., 2020), takes a high spatial resolution multispectral image and a low spatial resolution hyperspectral image as inputs and generates a high-resolution hyperspectral image. In this paper, we focus on the single image-to-image translation problem and leave this task as the future work.

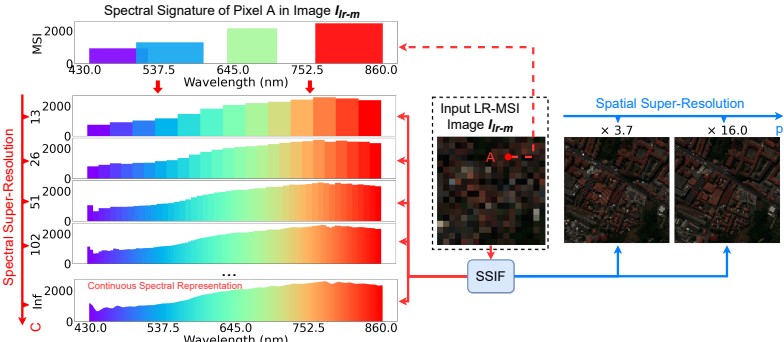

Figure 1: Spatial-Spectral Implicit Function (SSIF). Given an input low-resolution multispectral (LR-MSI) image, SSIF can perform both spatial (blue arrows) and spectral (red arrows) super-resolution simultaneously (illustrated with a specific pixel A). Unlike all the other neural implicit functions SSIF can generate images with any number of bands including "Inf" – a continuous function.

increasing the spatial resolution of the input images (e.g., from $h \times w$ pixels to $H \times W$ pixels) while keeping the spectral resolution (*i.e.*, number of spectral bands/channels) unchanged. In contrast, spectral SR (Galliani et al., 2017; Zhang, 2021) focuses on increasing the spectral resolution of the input images (e.g., from $c$ to $C$ channels) while keeping the spatial resolution fixed. SSSR (Mei et al., 2020; Ma et al., 2021) focuses on increasing both the spatial and spectral resolution of the input images. Here, $h, w$ (or $H, W$) indicates the height and width of the low-resolution, LR, (or high-resolution, HR) images while $c$ and $C$ indicates the number of bands/channels of the low/high spectral resolution images. For video signal, SR can also be done along the time dimension, but we don't consider it here and leave it as future work.

The diversity in input-output image resolutions (both spatial and spectral) significantly increases the complexity of developing deep neural network (DNN)–based SR models. Instead of jointly learning representations from images with different spatial and spectral resolutions, most SR research develops separate DNN models for each input-output image resolution pairs with a specific spatial and spectral resolution (Lim et al., 2017; Zhang et al., 2018b; Ma et al., 2021; Mei et al., 2020). For example, convolution-based SR models such as RCAN (Zhang et al., 2018a), SR3(Saharia et al., 2021), SSJSR (Mei et al., 2020) and (He et al., 2021) need to be trained separately for each input-output image resolution settings[3]. This practice has two limitations: 1) For some SR settings with much less training data, these models can yield suboptimal results or lead to overfitting; 2) It prevents generalizing trained SR models to unseen spatial/spectral resolutions.

Inspired by the recent progress in 3D reconstruction with implicit neural representation (Park et al., 2019; Mescheder et al., 2019; Chen & Zhang, 2019; Sitzmann et al., 2020; Mildenhall et al., 2020), image neural implicit functions (NIF) (Dupont et al., 2021; Chen et al., 2021; Yang et al., 2021; Zhang, 2021) partially overcome the aforementioned problems (especially the second one) by learning a continuous function that maps an arbitrary pixel spatial coordinate to the corresponding visual signal value; so in principle, they can generate images at any spatial resolution. For example, LIIF (Chen et al., 2021) is capable of generating images at any arbitrary resolution in the spatial domain. We call them *Spatial Implicit Functions (SIF)*. However, all current implicit function representations only focus on generalization in the spatial domain, and each SIF model is trained separately to target a specific spectral resolution (i.e., a fixed number of spectral bands).

In this work, we propose Spatial-Spectral Implicit Function $(SSIF)$, which generalizes the idea of neural implicit representations to the spectral domain. $SSIF$ represents an image as a continuous function on both pixel spatial coordinates in the spatial domain and wavelengths in the spectral domain. As shown in Figure 1, given an input low-resolution multispectral (or RGB) image , a single $SSIF$ model can generate images with different spatial resolutions and spectral resolutions. Note that extending the idea of implicit representations to the spectral domain is a non-trivial task. LIIF and other NIF models have an equal distance assumption in the spatial domain, meaning that pixels in the target HR image are assumed to be equally spaced. However, this equal distance assumption does not necessarily hold in the spectral domain. For many RGB or multispectral images, each band

---

[3]Figure 5a in Appendix A.1 illustrates this separate training practice.

may have different spectral widths, i.e., wavelength intervals of different lengths. Moreover, the wavelength intervals of different bands may overlap with each other. The "Spectral Signature of Pixel A" of the image $\mathbf{I}_{lr-m}$ in Figure 1 shows one example of such cases. To tackle this problem, we predict each spectral band value of each target pixel separately as the integral of the correlation between the pixel's radiance function and the current band's spectral response function over the desired spectral interval. Our contributions are as follows:

1. We propose Spatial-Spectral Implicit Function ($SSIF$) which represents an image as a continuous function on both pixel coordinates in the spatial domain and wavelengths in the spectral domain. $SSIF$ can handle SR tasks with different spatial and spectral resolutions simultaneously.

2. We demonstrate the effectiveness of $SSIF$ on two challenging spatio-spectral super-resolution benchmarks – CAVE (the indoor scenes) and Pavia Centre (Hyperspectral Remote Sensing images). We show that SSIF consistently outperforms state-of-the-art SR baseline models even when the baselines are trained separately at each spectral resolution (and spatial resolution), thus solving an easier task. Moreover, SSIF generalizes well to both unseen spatial resolutions and spectral resolutions.

3. We test the fidelity of the generated high resolution images on the downstream task of land use classification. Compared with the baselines, the images generated by $SSIF$ have much higher classification accuracy with 1.7%-7% performance improvements.

## 2 RELATED WORK

**Multispectral and Hyperspectral Image Super-Resolution**   As an ill-posed single image-to-image translation problem, super-resolution (SR) aims at increasing the spatial or spectral resolution of a given image such that it can be used for different downstream tasks. It has been widely used on natural images(Zhang et al., 2018a; Hu et al., 2019; Zhang et al., 2020b; Saharia et al., 2021; Chen et al., 2021), screen-shot images (Yang et al., 2021), omnidirectional images (Deng et al., 2021; Yoon et al., 2021) medical images (Isaac & Kulkarni, 2015), as well as multispectral (He et al., 2021) and hyperspectral remote sensing images(Mei et al., 2017; Ma et al., 2021; Mei et al., 2020; Wang et al., 2022b). It can be classified into three categories: spatial SR, spectral SR, and spatiospectral SR (SSSR). In this work, we focus on the most challenging task, SSSR, which subsumes spatial SR and spectral SR.

**Implicit Neural Representation**   Recently, we have witnessed an increasing amount of work using implicit neural representations for different tasks such as image regression (Tancik et al., 2020) and compression(Dupont et al., 2021; Strümpler et al., 2021), 3D shape regression/reconstruction (Mescheder et al., 2019; Tancik et al., 2020; Chen & Zhang, 2019), 3D shape reconstruction via image synthesis (Mildenhall et al., 2020), 3D magnetic resonance imaging (MRI) reconstruction (Tancik et al., 2020), 3D protein reconstruction (Zhong et al., 2020), spatial feature distribution modeling (Mai et al., 2020b; 2022; 2023b), remote sensing image classification (Mai et al., 2023a), geographic question answering (Mai et al., 2020a), and etc.. The core idea is to learn a continuous function that maps spatial coordinates (e.g., pixel coordinates, 3D coordinates, and geographic coordinates) to the corresponding signals (e.g., point cloud intensity, MRI intensity, visual signals, etc.). A common setup is to input the spatial coordinates in a deterministic or learnable Fourier feature mapping layer (Tancik et al., 2020) (consisting of sinusoidal functions with different frequencies), which converts the coordinates into multi-scale features. Then a multi-layer perceptron takes this multi-scale feature as input and whose output is used for downstream tasks. In parallel, implicit neural functions (INF) such as LIIF (Chen et al., 2021), ITSRN (Yang et al., 2021), Zhang (2021) are proposed for image super-resolution which map pixel spatial coordinates to the visual signals in the high spatial resolution images. One outstanding advantage is that they can jointly handle SR tasks at an arbitrary spatial scale. However, all the existing implicit functions learn continuous image representations in the spatial domain while still operating at fixed, pre-defined spectral resolutions. Our proposed SSIF overcomes this problem and generalizes INF to both spatial and spectral domains.

## 3 PROBLEM STATEMENT

The spatial-spectral image super-resolution (SSSR) problem over various spatial and spectral resolutions can be conceptualized as follows. Given an input low spatial/spectral resolution (LR-MSI) image $\mathbf{I}_{lr-m} \in \mathbb{R}^{h \times w \times c}$, we want to generate a high spatial/spectral resolution (HR-HSI) image $\mathbf{I}_{hr-h} \in \mathbb{R}^{H \times W \times C}$. Here, $h, w, c$ and $H, W, C$ are the height, width and channel dimension of

image $\mathbf{I}_{lr-m}$ and $\mathbf{I}_{hr-h}$, and $H > h$, $W > w$, $C > c$. The spatial upsampling scale $p$ is defined as $p = H/h = W/w$. Without loss of generality, let $\Lambda_{hr-h} = [\Lambda_0^T, \Lambda_1^T, ..., \Lambda_C^T] \in \mathbb{R}^{C \times 2}$ be the wavelength interval matrix, which defines the spectral bands in the target HR-HSI image $\mathbf{I}_{hr-h}$. Here, $\Lambda_i = [\lambda_{i,s}, \lambda_{i,e}] \in \mathbb{R}^2$ is the wavelength interval for the $i$th band of $\mathbf{I}_{hr-h}$ where $\lambda_{i,s}, \lambda_{i,e}$ are the start and end wavelength of this band. $\Lambda_{hr-h}$ can be used to fully express the spectral resolution of the target HR-HSI image $\mathbf{I}_{hr-h}$. In this work, we do not use $C/c$ to represent the spectral upsampling scale because bands/channels of image $\mathbf{I}_{lr-m}$ and $\mathbf{I}_{hr-h}$ might not be equally spaced (See Figure 1). So $\Lambda_{hr-h}$ is a very flexible representation for the spectral resolution, capable of representing situations when different bands have different spectral widths or their wavelength intervals overlap with each other. When $\mathbf{I}_{hr-h}$ has equally spaced wavelength intervals, such as those of most of the hyperspectral images, we use its band number $C$ to represent the spectral scale.

The spatial-spectral super-resolution (SSSR) can be represented as a function

$$\mathbf{I}_{hr-h} = H_{sr}(\mathbf{I}_{lr-m}, p, \Lambda_{hr-h}) \tag{1}$$

where $H_{sr}(\cdot)$ takes as input the image $\mathbf{I}_{lr-m}$, the desired spatial upsampling scale $p$, and the target sensor wavelength interval matrix $\Lambda_{hr-h}$, and generates the HR-HSI image $\mathbf{I}_{hr-h} \in \mathbb{R}^{H \times W \times C}$. In other words, we aim at learning **one single function** $H_{sr}(\cdot)$ that can take any input images $\mathbf{I}_{lr-m}$ with a fixed spatial and spectral resolution, and generate images $\mathbf{I}_{hr-h}$ with diverse spatial and spectral resolutions specified by different $p$ and $\Lambda_{hr-h}$.

Note that none of the existing SR models can achieve this. Most classic SR models have to learn separate $H_{sr}(\cdot)$ for different pairs of $p$ and $\Lambda_{hr-h}$ such as RCAN (Zhang et al., 2018a), SR3(Saharia et al., 2021), SSJSR (Mei et al., 2020), He et al. (2021). As for Spatial Implicit Functions (SIF) such as LIIF(Chen et al., 2021), SADN (Wu et al., 2021a), ITSRN (Yang et al., 2021), Zhang (2021), they can learn one $H_{sr}(\cdot)$ for different $p$ but with a fixed $\Lambda_{hr-h}$.

## 4 SPATIAL-SPECTRAL IMPLICIT FUNCTION

### 4.1 SENSOR PRINCIPLES

To design SSIF, we follow the physical principles of spectral imaging. Let $\mathbf{s}_{l,i}$ be the pixel density value of a pixel $\mathbf{x}_l$ at the spectral band $b_i$ with wavelength interval $\Lambda_i$. It can be computed by an integral of the **radiance function** $\gamma_{\mathbf{I}}(\mathbf{x}_l, \lambda)$ and **response function** $\rho_i(\lambda)$ of a sensor at band $b_i$.

$$\mathbf{s}_{l,i} = \int_{\Lambda_i} \rho_i(\lambda) \gamma_{\mathbf{I}}(\mathbf{x}_l, \lambda) \, \mathrm{d}\lambda \tag{2}$$

where $\lambda$ is wavelength. So for each pixel $\mathbf{x}_l$, the radiance function is a neural field that describes the radiance curve as a function of the wavelength. Note that unlike recent NeRF where only three discrete wavelength intervals (i.e., RGB) are considered, we aim to learn a *continuous* radiance curve for each pixel. The spectral response function (Zheng et al., 2020) describes the sensitivity of the sensor to different wavelengths and is usually sensor-specific. For example, the red sensor in commercial RGB cameras has a strong response (i.e., high pixel density) to red light. The spectral response functions of many commercial hyperspectral sensors (e.g., AVIRIS's ROSIS-03[4], EO-1 Hyperion) are very complex due to atmospheric absorption. A common practice adopted by many studies (Barry et al., 2002; Brazile et al., 2008; Cundill et al., 2015; Crawford et al., 2019; Chi et al., 2021) is to approximate the response function of individual spectral bands as a Gaussian distribution or a uniform distribution. In this work, we adopt this practice and show that this inductive bias enforced via physical laws improves generalization.

In the following, we will discuss the design of our SSIF which allows us to train a single SR model for different $p$ and $\Lambda_{hr-h}$. The whole model architecture of SSIF is illustrated in Figure 2b.

### 4.2 SSIF ARCHITECTURE

Following previous SIF works (Chen et al., 2021; Yang et al., 2021), SSIF first uses an image encoder $E_I(\cdot)$ to convert the input image $\mathbf{I}_{lr-m} \in \mathbb{R}^{h \times w \times c}$ into a 2D feature map $\mathbf{S}_{lr-m} = E_I(\mathbf{I}_{lr-m}) \in \mathbb{R}^{h \times w \times d_I}$ which shares the same spatial shape as $\mathbf{I}_{lr-m}$ but with a larger channel

---

[4]https://crs.hi.is/?page_id=877

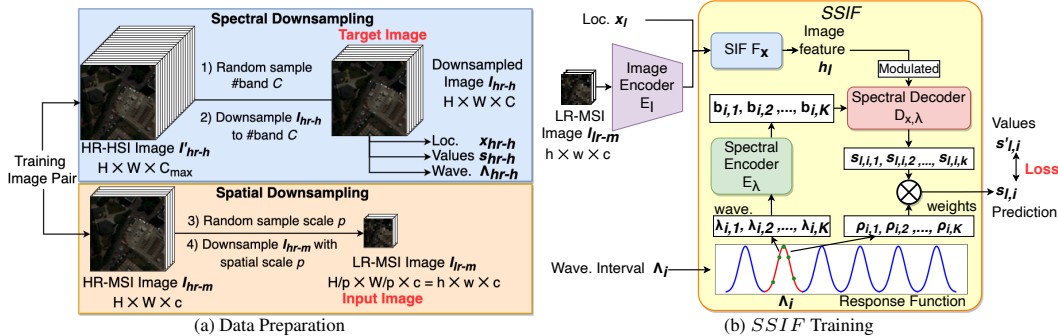

(a) Data Preparation               (b) $SSIF$ Training

Figure 2: Data preparation (a) and training (b) for $SSIF$. In Figure (b), we use Gaussian distributions as the response functions for different wavelength intervals $\{\Lambda_1, \Lambda_2, .., \Lambda_C\}$ while the response function $\rho_i(\lambda_{i,k})$ for $\Lambda_i$ is highlighted in red. The green dots are $K$ wavelengths $\{\lambda_{i,1}, \lambda_{i,2}, ..., \lambda_{i,K}\}$ sampled from a wavelength interval $\Lambda_i = [\lambda_{i,s}, \lambda_{i,e}] \in \Lambda_{hr-h}$ and $\{\rho_{i,1}, \rho_{i,2}, ..., \rho_{i,K}\}$ are their corresponding response function values. $\{\mathbf{b}_{i,1}, \mathbf{b}_{i,2}, ..., \mathbf{b}_{i,K}\}$ are their encoded spectral embeddings. $\bigotimes$ indicates dot product as shown in Equation 6.

dimension. $E_I(\cdot)$ can be any convolution-based image encoder such as EDSR (Lim et al., 2017) or RDN (Zhang et al., 2018b).

SSIF approximates the mathematical integral shown in Equation 2 as a weighted sum over the predicted radiance values of $K$ wavelengths $\{\lambda_{i,1}, \lambda_{i,2}, ..., \lambda_{i,K}\}$ sampled from a wavelength interval $\Lambda_i = [\lambda_{i,s}, \lambda_{i,e}] \in \Lambda_{hr-h}$ at location $\mathbf{x}_l$ (see Equation 3). Here, $\rho_i(\lambda_{i,k})$ is the response function value, i.e., weight, of each wavelength $\lambda_{i,k}$ given the current response function for band $b_i$. $\gamma_{\mathbf{I}}(\mathbf{x}_l, \lambda_{i,k})$ is the radiance value of $\lambda_{i,k}$ at location $\mathbf{x}_l$ which can be computed by a neural implicit function $G_{x,\lambda}$. Basically, $G_{x,\lambda}$ maps an arbitrary pixel location $\mathbf{x}_{l\in[-1,1]} \odot [-1,1]$ of $\mathbf{I}_{hr-h}$ and a wavelength $\lambda_{i,k} \in \Lambda_i$ into the radiance value of the target image $\mathbf{I}_{hr-h}$ at the corresponding location and wavelength, i.e., $\gamma_{\mathbf{I}}(\mathbf{x}_l, \lambda_{i,k}) = G_{x,\lambda}(\mathbf{S}_{lr-m}, \mathbf{x}_l, \lambda_{i,k})$. Here, $\odot$ is the Cartesian product.

$$\mathbf{s}_{l,i} = \sum_{k=1}^{K} \rho_i(\lambda_{i,k}) \gamma_{\mathbf{I}}(\mathbf{x}_l, \lambda_{i,k}) = \sum_{k=1}^{K} \rho_i(\lambda_{i,k}) G_{x,\lambda}(\mathbf{S}_{lr-m}, \mathbf{x}_l, \lambda_{i,k}) \qquad (3)$$

$G_{x,\lambda}$ can be decomposed into three neural implicit functions – a pixel feature decoder $F_{\mathbf{x}}$, a spectral encoder $E_\lambda$, and a spectral decoder $D_{\mathbf{x},\lambda}$. The pixel feature decoder takes the 2D feature map of the input image $\mathbf{S}_{lr-m}$ as well as one arbitrary pixel location $\mathbf{x}_{l\in[-1,1]} \odot [-1,1]$ of $\mathbf{I}_{hr-h}$ and maps them to a pixel hidden feature $\mathbf{h}_l \in \mathbb{R}^d$ where $d$ is the hidden pixel feature dimension (see Equation 4). Here, $F_{\mathbf{x}}$ can be any spatial implicit function such as LIIF Chen et al. (2021) and ITSRN (Yang et al., 2021).

$$\mathbf{h}_l = F_{\mathbf{x}}(\mathbf{S}_{lr-m}, \mathbf{x}_l) \qquad (4)$$

The spectral encoder $E_\lambda(\lambda_{i,k})$ encodes a wavelength $\lambda_{i,k}$ sampled from any wavelength interval $\Lambda_i = [\lambda_{i,s}, \lambda_{i,e}] \in \Lambda_{hr-h}$ into a spectral embedding $\mathbf{b}_{i,k} \in \mathbb{R}^d$. We can implement $E_\lambda$ as any position encoder (Vaswani et al., 2017; Mai et al., 2020b). Please refer to Appendix A.2 for a detailed description.

$$\mathbf{b}_{i,k} = E_\lambda(\lambda_{i,k}) \qquad (5)$$

Finally, the spectral decoder $D_{\mathbf{x},\lambda}(\mathbf{b}_{i,k}; \mathbf{h}_l)$ is a multilayer perceptron whose weights are modulated by the image feature embedding $\mathbf{h}_l$. $D_{\mathbf{x},\lambda}$ maps the spectral embedding $\mathbf{b}_{i,k}$ into a radiance value of $\lambda_{i,k}$ at location $\mathbf{x}_l$, i.e., $\mathbf{s}_{l,i,k} = D_{\mathbf{x},\lambda}(\mathbf{b}_{i,k}; \mathbf{h}_l)$. So we have

$$\mathbf{s}_{l,i} = \sum_{k=1}^{K} \rho_i(\lambda_{i,k}) G_{x,\lambda}(\mathbf{S}_{lr-m}, \mathbf{x}_l, \lambda_{i,k}) = \sum_{k=1}^{K} \rho_i(\lambda_{i,k}) D_{\mathbf{x},\lambda}(\mathbf{b}_{i,k}; \mathbf{h}_l) = \sum_{k=1}^{K} \rho_i(\lambda_{i,k}) \mathbf{s}_{l,i,k} \qquad (6)$$

The response function $\rho_i(\lambda_{i,k})$ can be a learnable function or a predefined function based on the knowledge of the target HSI sensor. To make the learning easier, we pick a predefined function, e.g. a Gaussian distribution or a uniform distribution, for each band $b_i$ by following Chi et al. (2021).

Figure 2b illustrates the model architecture of SSIF. The prediction $\mathbf{s}_{l,i} \in \mathbb{R}^C$ is compared with the ground truth $\mathbf{s}'_{l,i}$. A L1 reconstruction loss is used:

$$\mathcal{L} = \sum_{(\mathbf{I}_{lr-m}, \mathbf{I}_{hr-h}) \in \mathcal{D}} \sum_{(\mathbf{x}_l, \mathbf{s}_{hr-h}, \Lambda_{hr-h}) \in \mathbf{I}_{hr-h}} \sum_{\Lambda_i \in \Lambda_{hr-h}} \| \mathbf{s}_{l,i} - \mathbf{s}'_{l,i} \|_1, \qquad (7)$$

where $\mathcal{D}$ indicates all the low-res and high-res image pairs for the SSSR task.

### 4.3 Super-Resolution Data Preparation

Figure 2a illustrates the data preparation process of SSIF. Given a training image pair which consists of a high spatial-spectral resolution image $\mathbf{I}'_{hr-h} \in \mathbb{R}^{H \times W \times C_{max}}$ and an image with high spatial resolution but low spectral resolution $\mathbf{I}_{hr-m} \in \mathbb{R}^{H \times W \times c}$, we perform downsampling in both the spectral domain and spatial domain. For the spectral downsampling process (the blue box in Figure 2a), we downsample $\mathbf{I}'_{hr-h}$ in the spectral domain to obtain $\mathbf{I}_{hr-h} \in \mathbb{R}^{H \times W \times C}$ where the band number $C$ is sampled uniformly between the min and max band number $C_{min}, C_{max}$. For the spatial downsampling (the orange box in Figure 2b), we spatially downsample $\mathbf{I}_{hr-m}$ into $\mathbf{I}_{lr-m} \in \mathbb{R}^{h \times w \times c}$ which serves as the input for $SSIF$. Here, the downsampling scale $p$ is sampled uniformly from the min and max spatial scale $p_{min}, p_{max}$. See Appendix A.3 for a detailed description.

## 5 Experiments

To test the effectiveness of the proposed SSIF, we evaluate it on two challenging spatial-spectral super-resolution benchmark datasets – the CAVE dataset (Yasuma et al., 2010b) and the Pavia Centre dataset[5]. Both datasets are widely used for super-resolution tasks on hyperspectral images. Please refer to Appendix A.5 for detailed description of both datasets.

### 5.1 Baselines and SSIF Model Variants

Compared with spatial SR and spectral SR, there has been much less work on spatiospectral super-resolution. So we mainly compare our model with 7 baselines: **RCAN + AWAN**, **AWAN + RCAN**, **AWAN + SSPSR**, **AWAN + SSPSR**, **RC/AW + MoG-DCN**, **RC/AW + MoG-DCN**, **SSJSR**, **US3RN**, and **LIIF**. Please refer to Appendix A.4 for a detailed description for each baseline. For the first 6 baselines, we have to train separate SR models for different spatial and spectral resolutions of the output images. LIIF can use one model to generate output images with different spatial resolutions. However, we still need to train separate models when the output image $\mathbf{I}_{hr-m}$ with different band numbers $C$. In contrast, our $SSIF$ model is able to handle different spatial and spectral resolutions with one model.

Based on the response functions we use (Gaussian or uniform) and the wavelength sampling methods, we have 4 SSIF variants: **SSIF-RF-GS**, **SSIF-RF-GF**, **SSIF-RF-US**, and **SSIF-RF-UF**. Both SSIF-RF-GS and SSIF-RF-GF uses a Gaussian distribution $\mathcal{N}(\mu_i, \sigma_i^2)$ as the response function for each band $b_i$ with wavelength interval $\Lambda_i = [\lambda_{i,s}, \lambda_{i,e}]$ where $\mu_i = \frac{\lambda_{i,s} + \lambda_{i,e}}{2}$ and $\sigma_i = \frac{\lambda_{i,e} - \lambda_{i,s}}{6}$. The difference is SSIF-RF-GS uses $\mathcal{N}(\mu_i, \sigma_i^2)$ to sample $K$ wavelengths from $\Lambda_i$ while SSIF-RF-GF uses fixed $K$ wavelengths with equal intervals in $\Lambda_i$. Similarly, Both SSIF-RF-US and SSIF-RF-UF uses a Uniform distribution $\mathcal{U}(\lambda_{i,s}, \lambda_{i,e})$ as the response function for each band $b_i$. SSIF-RF-US uses $\mathcal{U}(\lambda_{i,s}, \lambda_{i,e})$ to sample $K$ wavelengths for each $\Lambda_i$ while SSIF-RF-UF uses fixed $K$ wavelengths with equal intervals. We also consider 1 additional SSIF variant – **SSIF-M** which only uses band middle point $\mu_i = \frac{\lambda_{i,s} + \lambda_{i,e}}{2}$ for each wavelength, i.e., $K = 1$.

### 5.2 SSSR on the CAVE dataset

Table 1 shows the evaluation result of the SSSR task across different spatial scales $p$ on the original CAVE dataset with 31 bands. We use three evaluation metrics - PSNR, SSIM, and SAM which measure the quality of generated images from different perspectives. We evaluate different baselines as well as $SSIF$ under different spatial scales $p = \{2, 4, 8, 10, 12, 14\}$. Since $p_{min} = 1$ and $p_{max} = 8$, $p = \{2, 4, 8\}$ indicates "in-distribution" results while $p = \{10, 12, 14\}$ indicates "Out-of-distribution" results for $p$ not present to LIIF or $SSIF$ during training time. We can see that

1. All 5 $SSIF$ can outperform or are comparable to the 7 baselines across all tested spatial scales even if the first 6 baselines are trained separately on each $p$.
2. SSIF-RF-UF achieves the best or 2nd best results across all spatial scales and metrics.
3. A general pattern we can see across all spatial scales is that the order of the model performances is SSIF-RF-* > SSIF-M > LIIF > other six baselines.

More interesting results emerge when we compare the performance of different models on different spectral resolutions, i.e., different $C$. Figure 3a and 3b compare model performance under different $C$ with a fixed spatial scale ($p = 4$ and $p = 8$ respectively). We can see that

1. Both Figure 3a and 3b show that SSIF-RF-UF achieves the best performances in two spatial scales and three metrics on "in-distribution" spectral resolutions.

---

[5] http://www.ehu.eus/ccwintco/index.php/Hyperspectral_Remote_Sensing_Scenes

Table 1: The evaluation result of the image super-resolution task across different spatial scales $p$ on the original CAVE (Yasuma et al., 2010a) dataset with 31 bands. "In-distribution" and "Out-of-distribution" indicate whether the model has seen this spatial scale $p$ during training. This is only applicable to LIIF (Chen et al., 2021) and our different versions of $SSIF$ models. The performance of LIIF and SSIF across different $p$ are obtained from the same model while for other 6 baselines, we trained separated SR models for each spatial scale $p$. Except for LIIF, the performances of all the other 6 baselines are from (Ma et al., 2021). We highlight the best model for each setting in bold and underline the second-best model.

| Model | In-distribution | | | | | | | | |
|---|---|---|---|---|---|---|---|---|---|
| Scale $p$ | 2 | | | 4 | | | 8 | | |
| Metric | PSNR ↑ | SSIM ↑ | SAM ↓ | PSNR ↑ | SSIM ↑ | SAM ↓ | PSNR ↑ | SSIM ↑ | SAM ↓ |
| RCAN(Zhang et al., 2018a) + AWAN(Li et al., 2020) | 36.22 | 0.971 | 8.81 | 32.69 | 0.935 | 9.82 | 28.25 | 0.834 | 11.73 |
| AWAN(Li et al., 2020) + RCAN(Zhang et al., 2018a) | 36.09 | 0.969 | 8.42 | 31.44 | 0.916 | 9.24 | 27.77 | 0.837 | 12.39 |
| AWAN(Li et al., 2020) + SSPSR(Mei et al., 2020) | 36.16 | 0.969 | 8.49 | 32.34 | 0.928 | 9.25 | 28.19 | 0.860 | 10.97 |
| RC/AW+MoG-DCN(Dong et al., 2021) | 36.12 | 0.969 | 8.53 | 32.68 | 0.923 | 9.44 | 28.33 | 0.853 | 13.2 |
| SSJSR(Mei et al., 2020) | 35.51 | 0.970 | 7.67 | 30.9 | 0.916 | 9.3 | 27.3 | 0.844 | 9.28 |
| US3RN(Ma et al., 2021) | 36.18 | 0.972 | 7.43 | 32.9 | 0.942 | 7.91 | 28.81 | 0.887 | 9.02 |
| LIIF(Chen et al., 2021) | 35.38 | 0.970 | 7.26 | 32.57 | 0.941 | 7.67 | 29.36 | 0.884 | 8.37 |
| SSIF-M | 35.80 | 0.972 | 7.21 | 32.91 | 0.944 | **7.54** | 29.54 | 0.888 | 8.26 |
| SSIF-RF-GS | 36.29 | 0.972 | 7.35 | 33.11 | 0.945 | 7.75 | 29.77 | 0.891 | 8.30 |
| SSIF-RF-GF | 36.37 | 0.972 | 7.49 | 33.22 | 0.945 | 7.96 | 29.90 | 0.892 | 8.45 |
| SSIF-RF-US | 36.23 | 0.971 | 7.54 | 33.11 | 0.943 | 7.91 | 29.85 | 0.891 | 8.39 |
| SSIF-RF-UF | **36.45** | **0.973** | **7.18** | **33.38** | **0.946** | 7.55 | **29.93** | **0.893** | **8.16** |

| Model | Out-of-distrobution | | | | | | | | |
|---|---|---|---|---|---|---|---|---|---|
| Scale $p$ | 10 | | | 12 | | | 14 | | |
| Metric | PSNR ↑ | SSIM ↑ | SAM ↓ | PSNR ↑ | SSIM ↑ | SAM ↓ | PSNR ↑ | SSIM ↑ | SAM ↓ |
| RCAN(Zhang et al., 2018a) + AWAN(Li et al., 2020) | - | - | - | - | - | - | - | - | - |
| AWAN(Li et al., 2020) + RCAN(Zhang et al., 2018a) | - | - | - | - | - | - | - | - | - |
| AWAN(Li et al., 2020) + SSPSR(Mei et al., 2020) | - | - | - | - | - | - | - | - | - |
| RC/AW+MoG-DCN(Dong et al., 2021) | - | - | - | - | - | - | - | - | - |
| SSJSR(Mei et al., 2020) | - | - | - | - | - | - | - | - | - |
| US3RN(Ma et al., 2021) | - | - | - | - | - | - | - | - | - |
| LIIF(Chen et al., 2021) | 27.59 | 0.859 | 8.62 | 26.67 | 0.838 | 8.96 | 25.5 | 0.822 | **9.17** |
| SSIF-M | 27.94 | 0.865 | 8.54 | 26.82 | 0.843 | 8.90 | 25.44 | 0.824 | 9.35 |
| SSIF-RF-GS | 27.98 | 0.866 | 8.59 | 27.03 | 0.848 | 8.95 | 25.50 | 0.828 | 9.45 |
| SSIF-RF-GF | 28.05 | **0.869** | 8.56 | 26.96 | 0.847 | 8.94 | **25.67** | **0.830** | 9.34 |
| SSIF-RF-US | **28.19** | 0.868 | 8.54 | 27.16 | **0.849** | 8.88 | 25.54 | 0.829 | 9.36 |
| SSIF-RF-UF | 28.14 | **0.869** | **8.45** | **27.17** | 0.849 | **8.77** | 25.62 | **0.830** | 9.19 |

Figure 3: The evaluation result of the SSSR task across different $C$ on the CAVE (Yasuma et al., 2010a) dataset. Here, the x axis indicates the number of bands $C$ of $\mathbf{I}_{hr-h}$. (a) and (b) compare the performances of different models across different $C$ in two spatial scales $p = 4$ or $p = 8$. Since our $SSIF$ can generalize to different $p$ and $C$, the evaluation metrics of each $SSIF$ are generated by one trained model. In contrast, we trained separated LIIF models for different $C$. The gray area in these plots indicates "out-of-distribution" performance in which $SSIF$ are evaluated on $C$s which have not been used for training.

2. However, the performance of SSIF-RF-UF, SSIF-RF-GF, and SSIF-M drop significantly when $C > 31$ while the performances of SSIF-RF-US and SSIF-RF-GS keep nearly unchanged for $C > 31$. This is because the first three SSIF use a fixed set of wavelengths during training while SSIF-RF-US and SSIF-RF-GS also sample novel wavelengths for each forward pass. This makes these two models have higher generalizability in "out-of-distribution" spectral scales.

3. A general pattern can be observed is that the order of model performance is SSIF-RF-* > SSIF-M > LIIF > other six baselines.

Ablation studies on different designs of spectral decoder $D_{\mathbf{x},\lambda}$ can be seen in Appendix A.7.

Table 2: Image super-resolution on the original Pavia Centre (Yasuma et al., 2010a) dataset with 102 bands. We evaluate models across different spatial scales $p = \{2, 3, 4, 8, 10, 12, 14, 16\}$. "In-distribution" and "Out-of-distribution" have the same meaning as Table 1. The performance of LIIF and SSIF across different $p$ are obtained from the same model while other 6 baselines need to be trained separately or each $p$. Except for LIIF, the performances of all the other 6 baselines are from (Ma et al., 2021). $SSIF - M*$ and $SSIF - M$ treat each band as a point while other $SSIF$ models treat each band as an interval.

| Model | In-distribution | | | | | | | | | | | |
|---|---|---|---|---|---|---|---|---|---|---|---|---|
| Scale $p$ | 2 | | | 3 | | | 4 | | | 8 | | |
| Metric | PSNR ↑ | SSIM ↑ | SAM ↓ | PSNR ↑ | SSIM ↑ | SAM ↓ | PSNR ↑ | SSIM ↑ | SAM ↓ | PSNR ↑ | SSIM ↑ | SAM ↓ |
| RCAN(Zhang et al., 2018a) + AWAN(Li et al., 2020) | 34.23 | 0.932 | 4.38 | 29.67 | 0.829 | 5.60 | 27.60 | 0.732 | 6.63 | 23.91 | 0.496 | 8.45 |
| AWAN(Li et al., 2020) + RCAN(Zhang et al., 2018a) | 34.54 | 0.936 | 4.38 | 29.66 | 0.827 | 5.70 | 27.61 | 0.734 | 6.69 | 23.67 | 0.515 | 8.87 |
| AWAN(Li et al., 2020) + SSPSR(Mei et al., 2020) | 34.24 | 0.934 | 4.30 | 29.60 | 0.828 | 5.55 | 27.71 | 0.742 | 6.32 | 24.21 | 0.506 | 8.14 |
| RC/AW+MoG-DCN(Dong et al., 2021) | 34.01 | 0.929 | 4.91 | 29.77 | 0.833 | 5.53 | 27.59 | 0.734 | 6.66 | 23.92 | 0.528 | 8.44 |
| SSJSR(Mei et al., 2020) | 31.80 | 0.894 | 4.80 | 29.05 | 0.810 | 6.14 | 27.06 | 0.703 | 6.93 | 20.61 | 0.347 | 18.30 |
| US3RN(Ma et al., 2021) | **35.86** | 0.951 | 3.71 | 30.38 | 0.857 | 4.88 | 28.23 | 0.775 | 5.80 | 24.26 | 0.548 | 7.96 |
| LIIF(Chen et al., 2021) | 35.24 | 0.952 | 3.91 | 30.72 | 0.881 | 4.76 | 28.67 | 0.815 | 5.43 | 24.52 | 0.551 | 7.72 |
| SSIF-M | 35.48 | **0.954** | 3.86 | **30.91** | **0.888** | 4.69 | 28.76 | 0.820 | 5.37 | 24.61 | 0.571 | 7.63 |
| SSIF-RF-GS | 35.47 | **0.954** | 3.87 | 30.87 | 0.887 | 4.71 | **28.84** | **0.821** | 5.37 | 24.62 | 0.569 | 7.62 |
| SSIF-RF-GF | 35.49 | **0.954** | 3.85 | 30.85 | 0.886 | 4.70 | 28.81 | **0.821** | 5.35 | 24.64 | **0.572** | 7.61 |
| SSIF-RF-US | 35.47 | **0.954** | 3.86 | **30.91** | 0.886 | 4.69 | 28.81 | **0.821** | 5.36 | **24.66** | **0.572** | 7.62 |
| SSIF-RF-UF | 35.48 | **0.954** | 3.84 | 30.88 | 0.886 | **4.68** | 28.83 | **0.821** | 5.35 | 24.60 | 0.570 | **7.60** |
| Model | Out-of-distribution | | | | | | | | | | | |
| Scale $p$ | 10 | | | 12 | | | 14 | | | 16 | | |
| Metric | PSNR ↑ | SSIM ↑ | SAM ↓ | PSNR ↑ | SSIM ↑ | SAM ↓ | PSNR ↑ | SSIM ↑ | SAM ↓ | PSNR ↑ | SSIM ↑ | SAM ↓ |
| RCAN(Zhang et al., 2018a) + AWAN(Li et al., 2020) | - | - | - | - | - | - | - | - | - | - | - | - |
| AWAN(Li et al., 2020) + RCAN(Zhang et al., 2018a) | - | - | - | - | - | - | - | - | - | - | - | - |
| AWAN(Li et al., 2020) + SSPSR(Mei et al., 2020) | - | - | - | - | - | - | - | - | - | - | - | - |
| RC/AW+MoG-DCN(Dong et al., 2021) | - | - | - | - | - | - | - | - | - | - | - | - |
| SSJSR(Mei et al., 2020) | - | - | - | - | - | - | - | - | - | - | - | - |
| US3RN(Ma et al., 2021) | - | - | - | - | - | - | - | - | - | - | - | - |
| LIIF(Chen et al., 2021) | 23.50 | 0.453 | 8.53 | 22.86 | 0.407 | 9.14 | 22.30 | 0.359 | **9.78** | **22.10** | 0.345 | **9.91** |
| SSIF-M | **23.53** | 0.466 | 8.47 | 22.82 | 0.404 | 9.19 | 22.31 | 0.359 | 9.85 | 22.05 | 0.344 | 9.99 |
| SSIF-RF-GS | 23.45 | 0.460 | 8.50 | 22.90 | 0.412 | 9.08 | 22.25 | 0.356 | 9.85 | 22.01 | 0.342 | 9.99 |
| SSIF-RF-GF | **23.53** | 0.466 | 8.44 | 22.96 | 0.413 | 9.01 | 22.22 | 0.350 | 9.86 | **22.10** | 0.340 | 9.94 |
| SSIF-RF-US | 23.46 | 0.458 | 8.57 | **22.99** | **0.415** | 9.03 | **22.33** | **0.363** | 9.82 | **22.10** | **0.346** | 10.02 |
| SSIF-RF-UF | 23.52 | **0.468** | 8.50 | 22.79 | 0.401 | 9.17 | 22.26 | 0.360 | 9.85 | 22.01 | 0.343 | 10.04 |

## 5.3 SSSR on the Pavia Centre Remote Sensing dataset

Table 2 shows the evaluation results of the SSSR task across different spatial scales $p = \{2, 3, 4, 8, 10, 12, 14, 16\}$ on the original Pavia Centre dataset with 102 bands. The setup is the same as Table 1. We can see that

1. Except for $p = 2$, all SSIF can outperform all baselines on different spatial scales.

2. The performances of 4 SSIF-RF-* models are very similar across different spatial scales while SSIF-RF-US is the winner in most cases. They can outperform LIIF in most settings.

Figure 4a and 4b compare different models across different spectral resolutions, i.e., $C$ for a fixed spatial scale ($p = 4$ and $p = 8$ respectively). We can see that

1. The performances of 4 SSIF-RF-* models can outperform SSIF-M which is better than LIIF, and the other 6 baselines.

2. All 4 4 SSIF-RF-* show good generalizability for "out-of-distribution" spectral scales, especially when $C > 102$ while SSIF-M suffers from performance degradation.

The ablation studies on $K$ and the generated remote sensing images can be seen in Appendix A.8.

## 5.4 Land Use Classification on the Pavia Centre Dataset

To test the fidelity of the generated high spatial-spectral resolution images, we evaluate them on land use classification task. We train the state-of-the-art land use classification model, A2S2K-ResNet (Roy et al., 2020), on the training dataset of Pavia Centre and evaluate its performance on the testing area – both the ground truth HSI image as well as the generated images from LIIF and different SSIF models. Table 3 compares the performance of A2S2K-ResNet on different generated images across different spatial scales. We can see that although SSIF-M shows good performance on the SSSR task on both datasets, the generated images are less useful – the land use classification accuracy on its generated images is much worse than other models, even far behind LIIF. SSIF-RF-GS shows the best performance across different spatial scales and can outperform LIIF by 1.7%-7%. Please refer to Appendix A.9 for a detailed description of the dataset, model, training detailed.

**Discussions of what the spectral encoder learned**  To understand how the spectral encoder represents a given wavelength we plot each dimension of spectral embedding against the wavelength

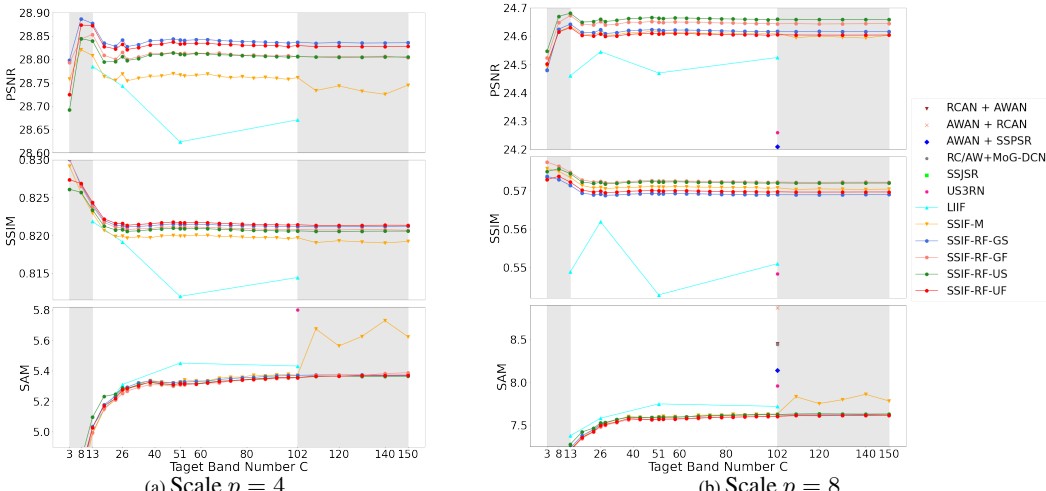

(a) Scale $p = 4$  (b) Scale $p = 8$

Figure 4: Evaluation across different $C$ on the Pavia Centre dataset. The set-up is the same as Figure 3. Note that some of the baseline models do not appear in some of those plots because the performances of these models are very low and cannot be shown in the current metric range.

Table 3: The evaluation of the generated images using A2S2K-ResNet (Roy et al., 2020) on the Pavia Centre land use classification task. "HSI" is the accuracy on the ground truth test image which is the upper bound. "Acc Imp." is the accuracy improvement from LIIF to SSIF-RF-GS.

| Model | Land Use Classification Accuracy (%) | | | |
|---|---|---|---|---|
| Band $C$ | 102 | | | |
| Scale $p$ | 2 | 3 | 4 | 8 |
| LIIF (Chen et al., 2021) | 41.69 | 41.29 | 37.87 | 37.38 |
| SSIF-M | 25.48 | 25.38 | 22.56 | 14.91 |
| SSIF-RF-GS | 43.44 | **46.86** | **44.97** | **44.82** |
| SSIF-RF-GF | 35.37 | 37.91 | 37.20 | 38.08 |
| SSIF-RF-US | 40.15 | 38.48 | 34.86 | 30.20 |
| SSIF-RF-UF | **45.32** | 44.00 | 41.87 | 36.34 |
| Acc Imp. | 1.75 | 5.57 | 7.10 | 7.44 |
| HSI (Upper Bound) | 72.66 | | | |

(Figure 10 in Appendix A.10). We find that they generally resemble piecewise-linear PL basis functions (Paul & Koch, 1974) or the continuous PK basis functions (Melal, 1976). This makes sense because PL and PK are classical methods to represent a scalar function – i.e., $G_{x,\lambda}(\mathbf{S}_{lr-m}, \mathbf{x}_l, \cdot)$ in our case. We can think that the weights of these basis are provided by the image encoder and SIF network given an image $\mathbf{S}_{lr-m}$ and location $\mathbf{x}_l$. Having a spectral encoder with learnable parameters should provide better representation than fixed basis functions.

## 6 CONCLUSION

In this work, we propose Spatial-Spectral Implicit Function (SSIF), a neural implicit model that represents an image as a continuous function of both pixel coordinates in the spatial domain and wavelengths in the spectral domain. This enables SSIF to handle SSSR tasks with different output spatial and spectral resolutions simultaneously with one model. In contrast, all previous works have to train separate super-resolution models for different spectral resolutions.

We demonstrate the effectiveness of SSIF on the SSSR task with Two datasets – CAVE and Pavia Centre. We show that SSIF can outperform all baselines across different spatial and spectral scales even when the baselines are allowed to be trained separately at each spectral resolution, thus solving an easier task. We demonstrate that SSIF generalizes well to unseen spatial and spectral resolutions. In addition, we test the fidelity of the generated images on a downstream task – land use classification. We show that SSIF can outperform LIIF with a big margin (1.7-7%).

In the current study, the effectiveness of SSIF is mainly shown on hyperspectral image SR, while SSIF is flexible enough to handle multispectral images with irregular wavelength intervals. This will be studied in future work. Moreover, the data limitation of the hyperspectral images poses a significant challenge to SR model training. We also plan to construct a large dataset for hyperspectral image super-resolution.

**Ethics Statement**    All datasets we use in this work including the CAVE and Pavia Centra datasets are publicly available datasets. No human subject study is conducted in this work. We do not find specific negative societal impacts of this work.

**Reproducibility Statement**    Our source code has been uploaded as a supplementary file to reproduce our experimental results. The implementation details of the spectral encoder are described in Appendix A.2. The SSIF model training details are described in Appendix A.6.

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

# A APPENDIX

## A.1 A ILLUSTRATION OF USING SSIF FOR MULTITASK IMAGE SUPER-RESOLUTION

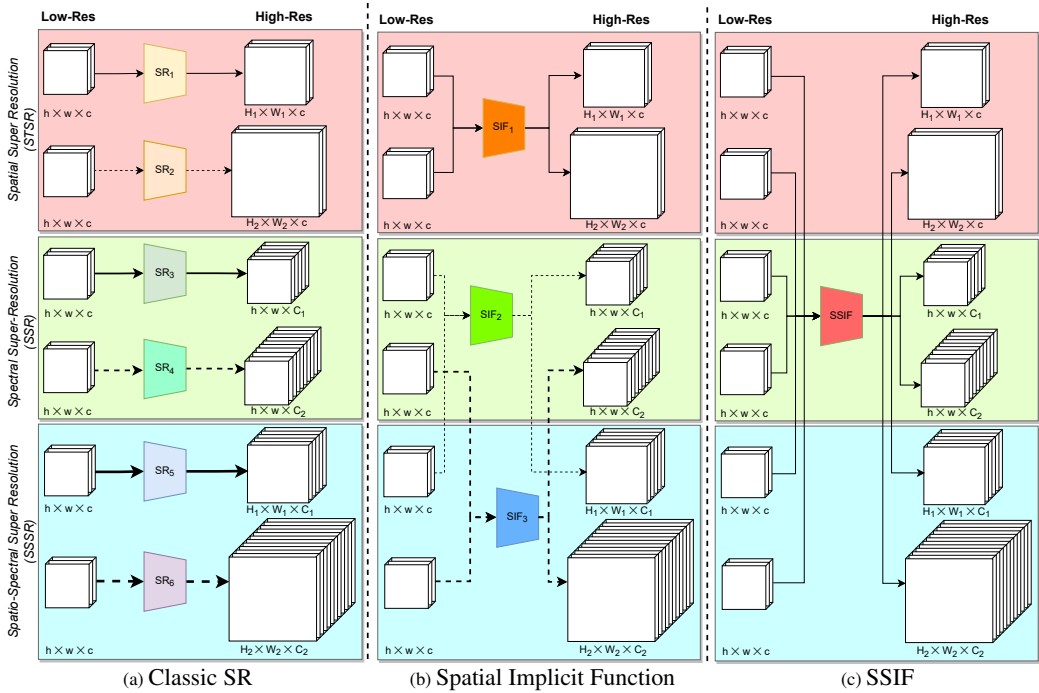

Figure 5: An illustration of image super-resolution on different spatial and spectral resolutions. The red, green, and blue boxes indicates three different super-resolution problems: Spatial Super-Resolution (spatial SR), Spectral Super-Resolution (spectral SR), and Spatio-Spectral Super-Resolution (SSSR). The three subfigures illustrate how the classic super-resolution models, the spatial implicit functions, and SSIF handle different SR tasks which generated image with different spatial and spectral resolutions. (a) Classic SR - most super-resolution models train **separate SR models** for different input and output image pairs with different spatial and spectral resolutions such as RCAN Zhang et al. (2018a), SR3Saharia et al. (2021), SSJSR Mei et al. (2020), He et al. (2021); (b) Spatial Implicit Function (SIF) - recently many research focus on using the idea of neural implicit function to develop spatial scale-agnostic super-resolution models such that one model can be trained to do super-resolution for different spatial scale such as MetaSRHu et al. (2019), LIIFChen et al. (2021), SADN Wu et al. (2021a), ITSRN Yang et al. (2021), Zhang (2021). However, they have to train separate SR models if target images have different spectral resolutions. (c) Spatial-Spectral Implicit Function ($SSIF$) aims at using one model to handle different spatial scales and spectral scales at the same time such that we can train one generic model for different SR tasks.

## A.2 SPECTRAL ENCODER $E_\lambda$

A key component of $SSIF$ is the spectral encoder $E_\lambda$ component. It consists of a Fourier feature mapping layer $\Psi(\cdot)$ followed by a multi-layer perceptron $MLP(\cdot)$:

$$\mathbf{b}_{i,k} = E_\lambda(\lambda_{i,k}) = MLP(\Psi(\lambda_{i,k})) \tag{8}$$

The Fourier feature mapping layer $\Psi(\cdot)$ takes a wavelength $\lambda_{i,k}$ sampled from the wavelength interval $\Lambda_i = [\lambda_{i,s}, \lambda_{i,e}] \in \Lambda_{hr-h}$ as the input and map it to a high dimensional vector $\mathbf{b}_{i,k} \in \mathbb{R}^d$, by using sinusoid functions with different frequencies. The idea is similar to the position encoder in Transformer (Vaswani et al., 2017), NeRF (Mildenhall et al., 2020), Space2Vec (Mai et al., 2020b; Tancik et al., 2020), and spatial implicit functions (Zhang, 2021; Dupont et al., 2021) for pixel location encoding. Here, we adopt the Space2Vec (Mai et al., 2020b) style position encoder $\Psi(\cdot)$. Let $\lambda_{min}, \lambda_{max}$ be the minimum and maximum scaling factor in the wavelength space, and $g = \frac{\lambda_{max}}{\lambda_{min}}$. We define $\Psi(\cdot)$ as Equation 9). Here, $\bigcup_{t=0}^{T-1}$ indicates vector concatenation through different scales.

$$\Psi(\lambda) = \bigcup_{t=0}^{T-1} \Big[ \sin(\frac{\lambda}{\lambda_{min} \cdot g^{t/(T-1)}}), \cos(\frac{\lambda}{\lambda_{min} \cdot g^{t/(T-1)}}) \Big]; \tag{9}$$

## A.3 SUPER-RESOLUTION DATA PREPARATION

Figure 2a illustrates the data preparation process of SSIF. Given a training image pair which consists of a high spatial-spectral resolution image $\mathbf{I}'_{hr-h} \in \mathbb{R}^{H \times W \times C_{max}}$ and an image with high spatial resolution but low spectral resolution $\mathbf{I}_{hr-m} \in \mathbb{R}^{H \times W \times c}$, we perform downsampling in both the spectral domain and spatial domain.

For the spectral downsampling process (the blue box in Figure 2a), we randomly sample a band number $C \sim Uni(C_{min}, C_{max})$ from a uniform distribution between the minimum and maximum band number $C_{min}, C_{max} > 0$. We use $C$ to downsample $\mathbf{I}'_{hr-h}$ in the spectral domain which yield $\mathbf{I}_{hr-h} \in \mathbb{R}^{H \times W \times C}$. Then we convert $\mathbf{I}_{hr-h}$ into location-value-wavelength samples $(\mathbf{x}_l, \mathbf{s}_{hr-h}, \Lambda)$. $\mathbf{x}_l$ and $\Lambda$ serve as the input features while $\mathbf{s}_{hr-h}$ are the prediction target. Note that, here we can sample equally spaced wavelength intervals or irregular spaced wavelength intervals for the target HR-HSI images $\mathbf{I}_{hr-h}$ since SSIF is agnostic to this irregularity.

For the spatial downsampling (the orange box in Figure 2b), we randomly sample a spatial scale $p \sim Uni(p_{min}, p_{max})$ where $Uni(p_{min}, p_{max})$ is a uniform distribution between the minimum and maximum spatial scale $p_{min}, p_{max} > 0$. We use $p$ to spatially downsample $\mathbf{I}_{hr-m}$ into $\mathbf{I}_{lr-m} \in \mathbb{R}^{h \times w \times c}$ which serves as the input for $SSIF$. Here, $h = H/p$ and $w = W/p$.

Interestingly, when the spatial upsampling scale $p$ is fixed as 1, our SSIF is degraded to a spectral SR model. When the band $C$ is fixed as the same as the input band, i.e., $C = c$, SSIF is degraded to a spatial SR model. When we vary $C$ and $p$ during SSIF training, we allow the model to do spatial SR and spectral SR at different difficulty levels which helps it to learn a continuous representation both in the spatial and spectral domain.

## A.4 BASLINES

We consider 7 baselines in our SSSR task on two benchmark dataset:

1. **RCAN + AWAN** uses RCAN (Zhang et al., 2018a) for spatial SR and then AWAN (Li et al., 2020) for spectral SR in a sequential manner.

2. **AWAN + RCAN** simply reverses the order of RCAN and AWAN.

3. **AWAN + SSPSR** uses AWAN and SSPSR (Mei et al., 2020) for spectral SR and spatial SR.

4. **RC/AW + MoG-DCN** first separately uses RCAN (Zhang et al., 2018a) to do spatial SR to obtain HR-MSI images and uses AWAN (Li et al., 2020) to do spectral SR to obtain LR-HSI images, and then uses MoG-DCN (Dong et al., 2021) to do hyperspectral image fusion based on the previously generated HR-MSI and LR-HSI images.

5. **SSJSR** (Mei et al., 2020) uses a fully convolution-based deep neural network to do SSSR.

6. **US3RN** (Ma et al., 2021) uses a deep unfolding network to solve the SSSR problem with a close-form solution. It is the current state-of-the-art model for the SSSR task.

7. **LIIF** (Chen et al., 2021) is initially designed for spatial SR on multispectral data. We increase the output dimension of LIIF's final MLP to allow it to work on hyperspectral images.

## A.5 DATASET DESCRIPTION

The CAVE dataset (Yasuma et al., 2010b) consists of 32 indoor hyperspectral (HSI) images captured under controlled illumination. Each image has a spatial size of $512 \times 512$ and 31 spectral bands covering the wavelength from 400nm to 700nm. Each HSI image is associated with an RGB image with the same spatial size. There are a lot of study using the CAVE dataset for hyperspectral image super-resolution (Yao et al., 2020; Mei et al., 2020; Zhang et al., 2020c; Zhang, 2021; Han et al., 2021; Qu et al., 2021; Ma et al., 2021). However, these works focus on different SR tasks. In this work, we focus on the most challenging task – SSSR. The train/test split on the CAVE dataset varies from paper to paper. In order to keep a fair comparison to the previous study, we adopt the train/test split from US3RN (Ma et al., 2021), the lastest work on this dataset, and use the first 22 samples

as the training dataset and the rest 10 samples as testing. The limited number of samples pose a significant challenges on modeling training. So similar to the previous work (Ma et al., 2021; Chen et al., 2021), given a HR-HSI and HR-MSI image pair $(\mathbf{I}'_{hr-h}, \mathbf{I}_{hr-m})$, we first do random croping for a $64p \times 64p$ image patch from both images. Then $\mathbf{I}_{hr-m}$ is spatially downsampled to a $64 \times 64$ image patch which serves as the input LR-MSI image $\mathbf{I}_{lr-m}$. We choose $p_{min} = 1$ and $p_{max} = 8$ and $p_{min} = 1$ for spatial downsampling, $C_{min} = 8$ and $C_{max} = 31$ for spectral downsampling (See Section 4.3).

The Pavia Centre (PC) dataset is taken by ROSIS, a widely used hyperspectral sensor. The images were collected over the center area of Pavia, northern Italy, in 2001. It contains 102 spectral bands covering secptrum from 430nm to 860nm. Figure 1 shows the spectral signature of one pixel A when $C = 102$. It has $1095 \times 715$ effective pixels. Similarly, we also adopt the train/test split from US3RN (Ma et al., 2021) and crop the upper left $1024 \times 128$ pixels as the testing dataset and the rest for training. The PC dataset does not come with a multispectral image counterpart. So we adopt the practice of (Mei et al., 2020) to simulate the high-resolution multispectral (HR-MSI) image based on the sensor specification of the multispectral sensor of IKONOS. The resulted image has 4 bands which correspond to R, G, B, and NIR. Please see the MSI spectral signature in Figure 1 for reference. Same random cropping technique is used for PC. We choose $p_{min} = 1$ and $p_{max} = 8$ and $p_{min} = 1$ for spatial downsampling, $C_{min} = 13$ and $C_{max} = 102$ for spectral downsampling (See Section 4.3).

## A.6  SSIF IMPLEMENTATION AND TRAINING DETAILS

We use RDN(Zhang et al., 2018b) as the image encoder $E_I(\cdot)$ and we use LIIF (Chen et al., 2021) as the pixel feature decoder $F_\mathbf{x}(\cdot)$.

For both CAVE and Pavia Centre dataset, we first tune the learning rate $lr = \{5.e-5, 1.e-4, 2.e-4\}$. We find out the default learning rate used by LIIF $lr = 1.e-4\}$ works the best for both datasets.

Then we tune the hyperparameters of LIIF including the output image feature dimension for image encoder $E_I(\cdot) - d_I = \{64, 128, 256\}$, the input image size $h = w \in \{48, 64\}$, the hidden dimension of LIIF's multi-layer perceptron – $h_{LIIF} \in \{256, 512\}$. We find out $d_I = 64$, $h = w = 64$, and $h_{LIIF} = 256$ give us the best results of LIIF on CAVE while for the Pavia Centre, $d_I = 256$, $h = w = 64$, and $h_{LIIF} = 512$ yield the best results. In addition, we find out using multiple dataloaders with different input image sizes $h = w$ is especially useful for the Pavia Centre dataset. In our experiment, we use three different dataloaders with $\{16, 32, 64\}$ as their input image size.

After we get the best hyperparameter combination of LIIF, we directly use them for SSIFwithout tuning. And we only tune the newly added hyperparameters for SSIFincluding the hidden dimension $h_{SSIP} = \{512, 1024\}$ of $MLP(\cdot)$ in Equation 8 and the wavelength sampling number $K \in \{2, 4, 8, 16, 32, 48, 52, 64\}$. We find out $h_{SSIP} = 512$ and $K = 16$ are the best hyperparameter combination for the CAVE dataset and $h_{SSIP} = 1024$ and $K = 64$ is the best for the Pavia Centre dataset.

All experiments are conducted on a Linux server with 1 CUDA GPU of 12GB memory. We use the official implementations[6] of LIIF (Chen et al., 2021) and US3RN (Ma et al., 2021). We implement our SSIF in PyTorch and will be made publicly available upon acceptance.

## A.7  ADDITIONAL EXPERIMENT RESULTS ON THE CAVE DATASET

Figure 6 illustrates the results of our ablation studies on different designs of spectral decoder $D_{\mathbf{x},\lambda}$ on the CAVE dataset. Two SSIF models – SSIF-RF-GS and SSIF-RF-UF – are used. We test two spectral decoder $D_{\mathbf{x},\lambda}$ variants:

1. "**D**": $D_{\mathbf{x},\lambda}$ is a multilayer perception (MLP) which is modulated by the image feature embedding $\mathbf{b}_{i,k}$. $D_{\mathbf{x},\lambda}$ takes a spectral embedding $\mathbf{b}_{i,k}$ as the input and output the corresponding radiance value. When $D_{\mathbf{x},\lambda}$ is a one-layer MLP, this can be seen as the dot product between the input spectral embedding $\mathbf{b}_{i,k}$ and image feature embedding $\mathbf{b}_{i,k}$.

---

[6]The LIIF implementation is under BSD 3-Clause "New" or "Revised" License.

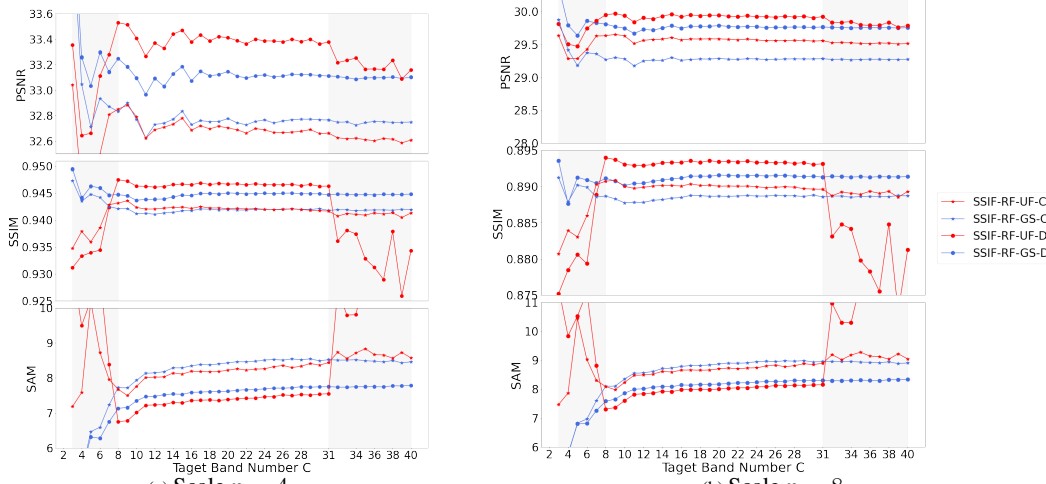

(a) Scale $p = 4$                    (b) Scale $p = 8$

Figure 6: The ablation studies of different designs of spectral decoder $D_{\mathbf{x},\lambda}$ on the CAVE dataset. Here, we use two SSIF models – SSIF-RF-GS (blue curves) and SSIF-RF-UF (red curves). Two spectral decoder $D_{\mathbf{x},\lambda}$ variants are explored: "**D**" and "**C**".

2. "**C**": $D_{\mathbf{x},\lambda}$ is a multilayer perception (MLP) which takes the concatenation of spectral embedding $\mathbf{b}_{i,k}$ and image feature embedding $\mathbf{b}_{i,k}$ as the input and output the corresponding radiance value.

Two SSIF models and two spectral decoder $D_{\mathbf{x},\lambda}$ variants amount to 4 different SSIF variants. From Figure 6, we can see that:

1. SSIF-RF-*-D usually outperform SSIF-RF-*-C which indicates that spectral decoder $D_{\mathbf{x},\lambda}$ variant **D** is usually more effective than **C**.

2. We find out SSIF-RF-UF-D can outperform SSIF-RF-UF-C for in-distribution spectral resolutions (i.e., $8 \le C \le 31$) while SSIF-RF-UF-C has better generalizability than SSIF-RF-UF-D for out-of-distribution spectral resolutions (i.e., $C > 31$).

## A.8    ADDITIONAL EXPERIMENTS RESULTS ON THE PAVIA CENTRE DATASET

We conduct the ablation study on the effect of the number of sampled wavelengths in each wavelength interval $\Lambda_i - K$ on the model performance. We use the Pavia Centra dataset as an example and compare model performances of SSIF-RF-US with different $K$. Figure 7 illustrates the results. We can see that a bigger $K$ leads to better model performance.

## A.9    DESCRIPTIONS OF THE LAND USE CLASSIFICATION TASK ON PAVIA CENTRE DATASET

**Motivation**    In addition to directly comparing the generated images with the ground truth images by using those image similarity metrics, many super-resolution works use human evaluation to evaluate whether the generated image looks natural or not (Khrulkov & Babenko, 2021; Saharia et al., 2021; He et al., 2021). However, as for super-resolution on remote sensing (RS) image datasets like Pavia Centre, the objective is not to generate RS image that look natural for human eyes but to generate RS images which can be useful for downstream tasks such as land use classification.

**Pavia Centre Land Use Classification Dataset**    So in this work, we conduct an additional evaluation on our SSIF  as well as the strongest baseline – LIIF by using land use classification task to test the fidlity of the generated RS images. The Pavia Centre dataset comes with a human annotated land use classification map with 10 different land use types for each pixel including water, trees, asphalt, self-blocking bricks, bitumen, tiles, shadows, meadows, and bare soil. For a statistic for each land use types, please refer to the original Pavia Centre webpage[7]. So we use it as the ground truth labels.

---

[7]http://www.ehu.eus/ccwintco/index.php/Hyperspectral_Remote_Sensing_Scenes

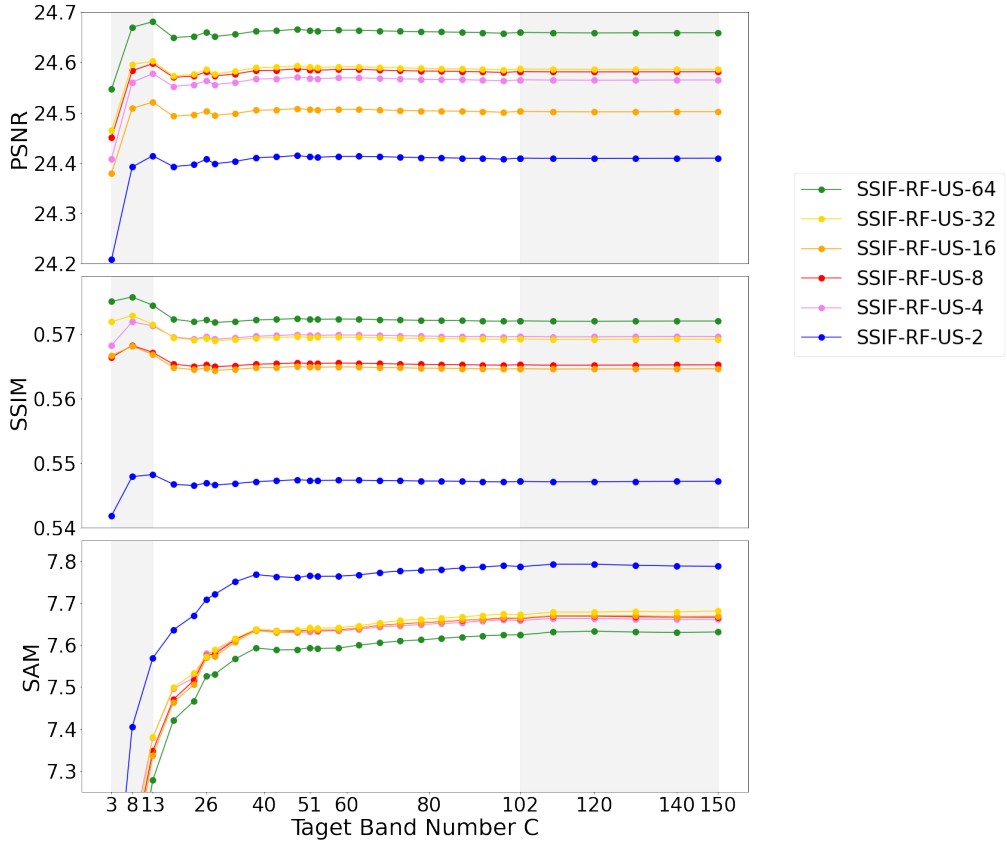

Figure 7: The ablation study on the number of sampled wavelengths in each wavelength interval $\Lambda_i - K$ on the Pavia Centra dataset with spatial scale $p = 8$. The setting is similar to Figure 4. We use SSIF-RF-US model as an example and tune the hyperparameter $K = \{2, 4, 6, 8, 16, 32, 64\}$. Here, each SSIF is named as SSIF-RF-US-K. We can see that a bigger $K$ leads to better model performance.

**Land Use Classification Model** In terms of the land use classification model, we need to select an appropriate image segmentation model for the Pavia Centre dataset. The recent Segmenter (Strudel et al., 2021) network extends the Vision Transformer to a semantic segmentation model which shows the state-of-the-art performance on multiple RGB image segmentation datasets. Ayush et al. (2021) utilized PSANet (Zhao et al., 2018) network with ResNet-50 backbone to perform semantic segmentation on the SpaceNet satellite image dataset (Van Etten et al., 2018). However, both models are designed for semantic segmentation on RBG images while what we want is an image segmentation model on hyperspectral images which contain hundreds of bands. SSJSR (Mei et al., 2020) chooses a simple per-pixel-based support vector machine (SVM) model to do image segmentation on hyperspectral RS images for the super-resolution model evaluation. However, SVM does not consider the spatial neighborhood of each pixel so it cannot take into account the spatial correlations among nearby pixels which will lead to suboptimal results. We finally choose to use A2S2K-ResNet (Roy et al., 2020) as our image segmentation model. A2S2K-ResNet is ranked the third place in the PaperWithCode leaderboard[8] on the Pavia University dataset which is a hyperspectral image classification/segmentation dataset. The hyperspectral images from the Pavia University dataset was taken by exactly the same ROSIS sensors as the Pavia Centre dataset. Moreover, both datasets were collected at relatively the same time and nearby locations. Since we cannot find a leaderboard for the Pavia Centre dataset, we choose to use the leaderboard of the Pavia University dataset for reference. Note that the first ranked model – SpectralNET (Chakraborty & Trehan, 2021) is from an unpublished ArXiv paper. And both the first and second-ranked model – SpectralNET (Chakraborty & Trehan, 2021) and SSDGL (Zhu et al., 2021) have a poorly organized codebase which causes difficulties for us to reproduce their results. So we choose the third-ranked model – A2S2K-ResNet (Roy et al., 2020) which is a modified ResNet model for hypersepctral image classification/segmentation.

---

[8]https://paperswithcode.com/sota/hyperspectral-image-classification-on-pavia

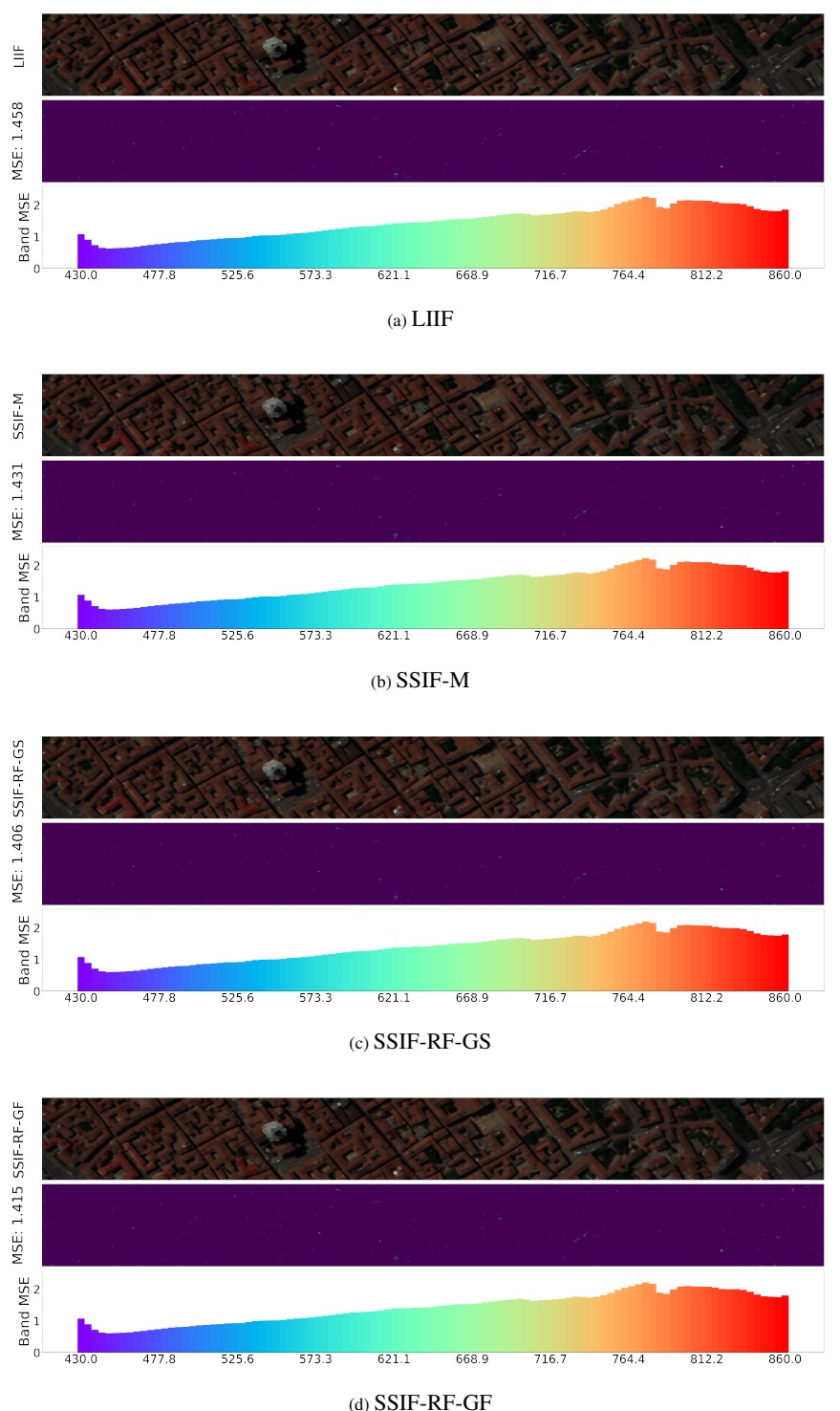

Figure 8: The comparison among the generated images from LIIF and different SSIF for $p = 4$ and $C = 102$. For Figure (a)-(g), we first show the generated image of the corresponding model. Then we show the MSE between it and the ground truth image per pixel. The y axis label indicates the MSE value which times 1000. Then finally, we show the MSE for each band. Figure (g) shows the ground truth image.

**Model Training Detail** We follow the exact training process of A2S2K-ResNet. Within the training region of the Pavia Centre hyperspectral image, we balanced sample 2000 pixel samples for each

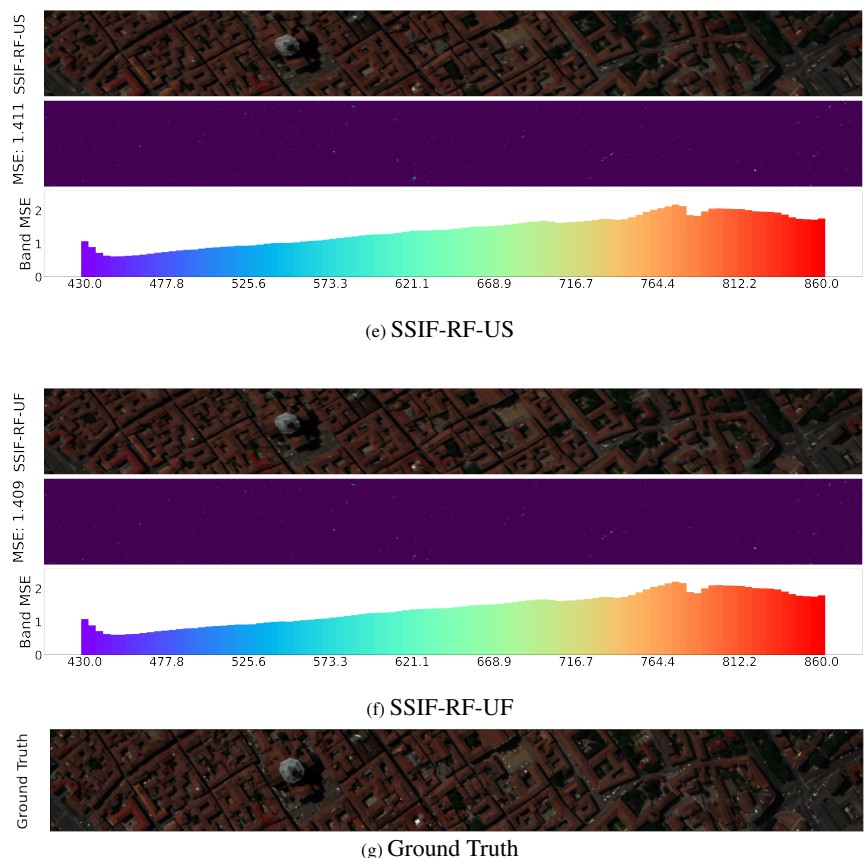

Figure 8: The comparison among the generated images from LIIF and different SSIF for $p = 4$ and $C = 102$. For Figure (a)-(g), we first show the generated image of the corresponding model. Then we show the MSE between it and the ground truth image per pixel. The y axis label indicates the MSE value which times 1000. Then finally, we show the MSE for each band. Figure (g) shows the ground truth image.

land use type. For each training pixel, we crop a $11 \times 11$ spatial neighborhood region as the input for A2S2K-ResNet. The model takes the $11 \times 11 \times 102$ input tensor and produces a probability distribution over all land use types to the current center pixel. In the test region, we also balanced sample 500 pixel samples for each land use type as the validation dataset. As reported in the Roy et al. (2020), We train A2S2K-ResNet for 200 epochs and take the one model instance that has the highest performance on our validation dataset. We use this trained A2S2K-ResNet model to produce the predicted land use classification maps based on either the ground truth HSI images or the generated images from LIIF or SSIF for the test region. And then, we obtain the classification accuracy on each image. The results are shown in Table 3.

## A.10 VISUALIZING THE BASIS IN SPECTRAL ENCODER

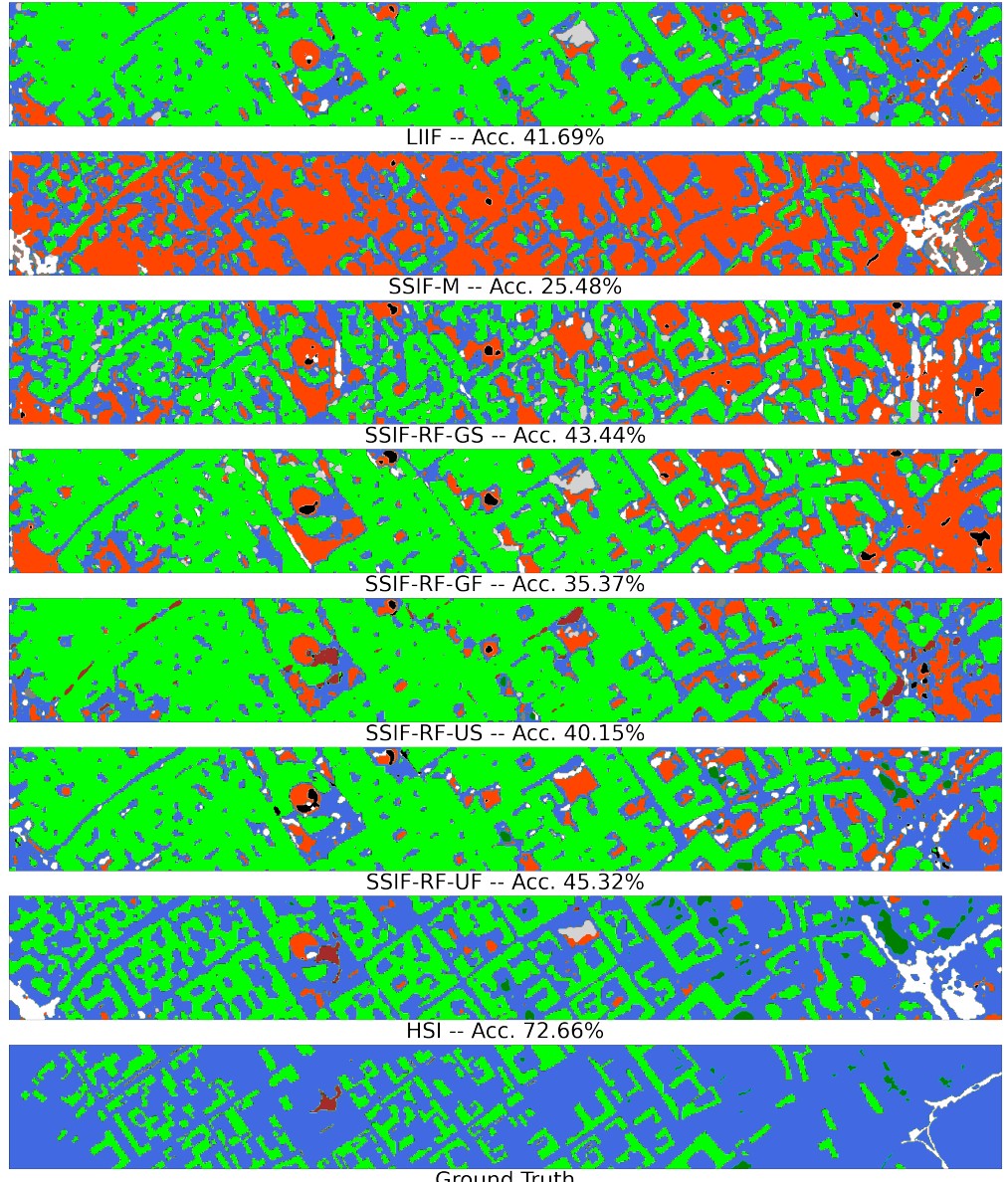

LIIF -- Acc. 41.69%

SSIF-M -- Acc. 25.48%

SSIF-RF-GS -- Acc. 43.44%

SSIF-RF-GF -- Acc. 35.37%

SSIF-RF-US -- Acc. 40.15%

SSIF-RF-UF -- Acc. 45.32%

HSI -- Acc. 72.66%

Ground Truth

Figure 9: The land use classification results of A2S2K-ResNet (Roy et al., 2020) on the generated images under spatial upsampling scale $p = 2$ from LIIF, SSIF-M, SSIF-RF-GS, SSIF-RF-GF, SSIF-RF-US, SSIF-RF-UF, as well as on the original Pavia Centre test image. "Ground Truth" indicates the ground truth labels. The classification accuracy is also listed under each figure.

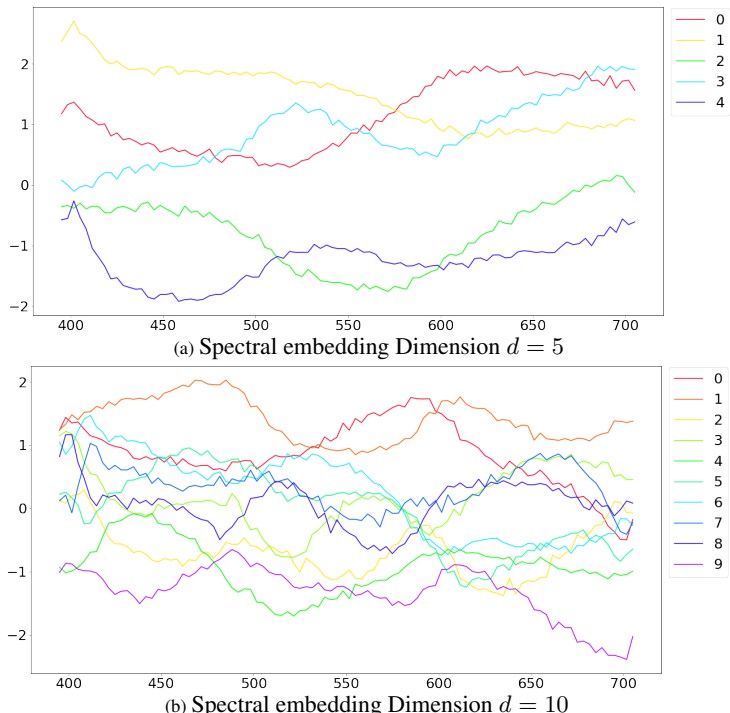

(a) Spectral embedding Dimension $d = 5$

(b) Spectral embedding Dimension $d = 10$

Figure 10: Visualizations of the spectral embeddings with small spectral embedding dimensions $d = \{5, 10\}$. Here we draw a curve for each dimension of the embedding, derived from the spectral encoders $E_\lambda$ of two learned SSIF-RF-GS. The x axis indicates the wavelength and each curve $E_\lambda(\lambda)[j]$ corresponds to the values of a specific spectral embedding dimension $j$.

