# OpenReview forum: "SSIF: Learning Continuous Image Representation for Spatial-Spectral Super-Resolution"
_ICLR.cc/2024/Conference — ICLR 2024 Conference Withdrawn Submission_

### Official Review · Reviewer_Hc3v · 2023-10-27

**Soundness:** 3 good
**Presentation:** 3 good
**Contribution:** 3 good
**Rating:** 6
**Confidence:** 5

**Summary:**

This work proposes Spatial-Spectral Implicit Function (SSIF), a neural implicit model that represents an image as a function of both continuous pixel coordinates in the spatial domain and continuous wavelengths in the spectral domain. SSIF can handle SR tasks with different spatial and spectral resolutions simultaneously.

**Strengths:**

(i) The idea of introducing INR to SSSR for rescaling spectral images with different spatio-spectral resolutions is interesting and novel. It is a good exploration in hyperspectral image domain since the arbitrary RGB image super-resolution has achieved great success. But the INR was mainly applied in the spatial dimension. Thus, jointly exploiting the INR technique in spatial and spectral dimension is a good attempt.

(ii) The writing is easy to follow and the presentation is well-dressed. The math notation in the method part is clear.

(iii) Code has been submitted. The reproducibility can be ensured.

(iv) The improvements of SSIF for the downstream tasks are significant, from 1.7% to over 7%

**Weaknesses:**

(i) The layout in some places is somewhat dense, please use the `vspace` command appropriately. For example, at the end of page 6, the caption of Fig. 3, etc.

(ii) There is no ablation study part to show the effectiveness of each component of the proposed SSIF. How SSIF was designed and which parts of it are most useful. These issues are yet to be clarified.

(iii) Lacks discussion and comparison with RGB arbitrary SR techniques. Since the INR technique has been widely applied for RGB arbitrary SR, it is better to discuss the differences between the proposed SSIF with them to make the contribution clearer. There are two works for example:

[1] VideoINR: Learning Video Implicit Neural Representation for Continuous Space-Time Super-Resolution. In CVPR 2022.
https://github.com/Picsart-AI-Research/VideoINR-Continuous-Space-Time-Super-Resolution

[2] Learning A Single Network for Scale-Arbitrary Super-Resolution. In ICCV 2021.
https://github.com/The-Learning-And-Vision-Atelier-LAVA/ArbSR

[3] Implicit Neural Representation Learning for Hyperspectral Image Super-Resolution. In TGRS 2023.

(iv) The baseline model used for SSSR is too old. AWAN was proposed three years ago. It has been significantly outperformed by a recent SOTA method MST [4] or MST++ [5], which won the first place of NTIRE 2022 spectral recovery from RGB images. Mean while this method costs less than 5% Params and FLOPS of AWAN. It is interesting to see the experiments of SSIF on MST or MST++

[4] Mask-guided Spectral-wise Transformer for Efficient Hyperspectral Image Reconstruction. In CVPR 2022.
https://github.com/caiyuanhao1998/MST/

[5] MST++: Multi-stage Spectral-wise Transformer for Efficient Spectral Reconstruction. In CVPRW 2022.
https://github.com/caiyuanhao1998/MST-plus-plus

**Questions:**

To your knowledge, is this work the first attempt to introduce INR into SSSR?

If so, it is a great exploration.

If not, it is better to discuss with previous similar method and make a comparison.

---

### Official Review · Reviewer_uUms · 2023-10-28

**Soundness:** 2 fair
**Presentation:** 2 fair
**Contribution:** 2 fair
**Rating:** 3
**Confidence:** 4

**Summary:**

In this paper, the authors establish continuous representatives for hyperspectral images in the spatial and spectral domains via neural implicit functions (NIFs), namely, Spatial-Spectral Implicit Function (SSIF). The implicit representations of spatial and spectral domain are decoupled into spatial and spectral implicit representation, respectively. SSIF provides the corresponding solutions for the two subtasks. SSIF can handle SR tasks with different spatial and spectral resolutions simultaneously.

**Strengths:**

1.	The proposed method can interpolate and extrapolate both spatial and spectral resolution of HSIs.
2.	Experiments on the classification task shows the application potential of SSIF.

**Weaknesses:**

1.	The authors reviewed many SR methods for natural images and pointed out that these methods did not consider the spectral domain. Actually, these methods all aim for RGB images, in which the spectrum are not concerned. Thus, it is unfair and unreasonable to state the motivations and contributions from this perspective.
2.	The proposed method aims to accomplish the spatial and spectral resolution for hyperspectral images. However, few works about hyperspectral image resolution are involved in the paper. In contrast, the authors reviewed lots of works for RGB images resolution. There is a gap between these two.
3.	The proposed method involves with the implicit representation of 3D cube data. The differences with 3D reconstruction and video super resolution should be clarified.
4.	In this paper, the task is decoupled into two subtasks: the implicit representation of spatial domain and spectral domain. There are many solutions for the former subtask. It is trivial. For the latter one, it seems that the latent codes for different sampling wavelengths in an interval are simply weighted by SRF.
5.	Although the paper aims to reconstruct high-resolution images without knowing the PSF and SRF, the training data is still generated with known PSF and SRF. This means the network may not work with unknown PSF and SRF. Is it possible to deploy the method on real data?
6.	The spectrum is extrapolated via the proposed method SSIF. Are the extrapolated HSIs with more spectrum more effective for downstream tasks?
7.	The comparison methods should contain the methods designed for HSI resolution instead of many methods for RGB images, which may not consider the data properties.
8.	The effectiveness for separate resolution tasks is not verified.
9.	There are many typos and some illustrations are low quality for identification, such as Figure 3 and 8.
10.	The reviewer concerns the IOU for each class and mIOU more.

**Questions:**

See above

---

### Official Review · Reviewer_jdfT · 2023-10-29

**Soundness:** 2 fair
**Presentation:** 2 fair
**Contribution:** 2 fair
**Rating:** 5
**Confidence:** 3

**Summary:**

This paper proposes a spatial-spectral implicit function, trying to continuously represent images in the spatial and spectral domains.

**Strengths:**

1. The authors propose the spatial-spectral implicit function (SSIF) to represent an image as a continuous function in the spatial and spectral domains.
2. The proposed SSIF outperforms SR baselines.

**Weaknesses:**

1.	The authors should give the reason why they introduce the implicit neural representation to spectral reconstruction.
2.	The authors claim that the idea of implicit representations to the spectral domain is a non-trivial task since existing methods have an equal distance assumption in the spatial domain. However, what is the assumption of the spectral image,  why does each band have different spectral widths?
3.	The authors ignore the works using implicit neural representation for spectral SR [4,5] and spatial-spectral [6], as well as the state-of-the-art spatial SR works using implicit neural representation [1,2,3].
4.	How to ensure the spatial-spectral correlation?
5.	The authors should give a visual comparison to verify the effectiveness of the proposed method.
6.	The authors should give the results of joint spatial-spectral SR for out-of-distribution.
7.	The authors use a 2D image encoder (EDSR) to extract features, which can only take the image with fixed channel numbers? However the data preparation generates input images with different channels, this is unclear.
[1] CiaoSR: Continuous Implicit Attention-in-Attention Network for Arbitrary-Scale Image Super-Resolution, CVPR2023.
[2] Cascaded Local Implicit Transformer for Arbitrary-Scale Super-Resolution, CVPR2023.
[3] Local Texture Estimator for Implicit Representation Function, CVPR2022.
[4] Implicit Neural Representation Learning for Hyperspectral Image Super-Resolution, ICME2022.
[5] Continuous Spectral Reconstruction from RGB Images via Implicit Neural Representation, ECCVW2022.
[6] Hyperspectral Image Joint Super-Resolution via Implicit Neural Representation, IMT2022.

**Questions:**

How to calculate the metric beyond 31 bands?
Do the authors conduct experiments on other spectral images, such as ICVL [1] and NTIRE2022 [2]?

[1] Sparse Recovery of Hyperspectral Signal from Natural RGB Images, ECCV2016.
[2] NTIRE 2022 Spectral Recovery Challenge and Dataset, CVPRW2022.

---

### Official Review · Reviewer_WHXm · 2023-10-29

**Soundness:** 2 fair
**Presentation:** 2 fair
**Contribution:** 2 fair
**Rating:** 3
**Confidence:** 5

**Summary:**

This paper proposes a neural implicit model called spatial-spectral implicit function for hyperspectral image spatial-spectral super-resolution, which represents an image as a function of both continuous pixel coordinates in the spatial domain and continuous wavelengths in the spectral domain. The experimental results show that the peroposed method outperforms the compared methods.

**Strengths:**

The proposed method has good generalization ability for unseen super-resolution scale and spectral wavelength, which is important for image restoration especially for hyperspectral image restoration with limited data.

**Weaknesses:**

1. Neural implicit model for image spatial super-resolution has been proposed in previous works, and extending it for spatial-spectral super-resolution is trivial. According to the experiments, the SSIF-RF-U* almost performs best, which means spectral super-resolution can follows the equal distance assumption.
2. The representation is hard to fellow. Some symbols are wrong, for example s_{l,i} \in R^C should be s_{l,i} \in R in page 5.
3. The font is inconsistent, such as NIF, SIF and SSIF.
4. Lacking visualization. As a image restoration task, some visualization should be provided in manuscript rather than in supplementary. Besides, the visualization in supplementary is hard to verify the effectiveness of the proposed method.
5. The results of RCAN+AWAN to US3RN in Tables 1 and 2 are redundance and take too much space.
6. The experiment of classification is meaningless. Compared with original hyperspectral image, the accuracy of super-resolved image degraes more than 20% and is lower 50%, which means the classification model is not work.

**Questions:**

I am curious about how to generat paired I'_{hr-h} and I_{hr-m}. In general, we only have I'_{hr-h}.